# Study on the Influence of the Preparation Method of Konjac Glucomannan-Silica Aerogels on the Microstructure, Thermal Insulation, and Flame-Retardant Properties

**DOI:** 10.3390/molecules28041691

**Published:** 2023-02-10

**Authors:** Ying Kuang, Puming Liu, Yichen Yang, Xiaosa Wang, Menglong Liu, Wei Wang, Tianlin Guo, Man Xiao, Kai Chen, Fatang Jiang, Cao Li

**Affiliations:** 1National “111” Center for Cellular Regulation and Molecular Pharmaceutics, Hubei Key Laboratory of Industry Microbiology, Glyn O. Phillips Hydrocolloid Research Center at HBUT, Hubei University of Technology, Wuhan 430068, China; 2Faculty of Engineering, University of Nottingham, Nottingham NG7 2RD, UK; 3School of Health Science and Engineering, Hubei University, Wuhan 430062, China

**Keywords:** Konjac glucomannan, silica aerogel, preparation method, microstructure, thermal insulation

## Abstract

Natural polysaccharides with high viscosity, good thermal stability, and biocompatibility can improve the mechanical properties of inorganic silica aerogels and enhance their application safety. However, the effects of the preparation methods of polysaccharide-silica aerogels on their microstructure and application properties have not been systematically studied. To better investigate the effect of the microstructure on the properties of aerogel materials, two aerogels with different structures were prepared using Konjac glucomannan (KGM) and tetraethoxysilane (TEOS) via physical blending (KTB) and co-precursor methods (KTC), respectively. The structural differences between the KTB and KTC aerogels were characterized, and the thermal insulation and fire-retardant properties were further investigated. The compressive strength of the KTC aerogels with a cross-linked interpenetrating network (IPN) structure was three times higher than that of the KTB aerogels, while their thermal conductivity was 1/3 of that of the KTB aerogels. The maximum limiting oxygen index (*LOI*) of the KTC aerogels was 1.4 times, the low peak heat release rate (PHRR) was reduced by 61.45%, and the lowest total heat release (THR) was reduced by 41.35% compared with the KTB aerogels. The results showed that the KTC aerogels with the IPN have better mechanical properties, thermal insulation, and fire-retardant properties than the simple physically blending KTB aerogels. This may be due to the stronger hydrogen-bonding interactions between KGM and silica molecules in the KTC aerogels under the unique forcing effect of the IPN, thus enhancing their structural stability and achieving complementary properties. This work will provide new ideas for the microstructure design of aerogels and the research of new thermal insulation and fire-retardant aerogels.

## 1. Introduction

Aerogels are a kind of ultra-lightweight porous solid material with a unique three-dimensional nanonetwork structure using the gas phase as the dispersion medium [1]. They have the characteristics of high porosity, low thermal conductivity, low density, and high specific surface area. Since Kistler prepared the first aerogel by catalyzing sodium silicate with hydrochloric acid [2], they have been widely used in adsorption materials, oil-water separation [3,4], catalysis, thermal and acoustic insulation [5], and other fields. Silica aerogels which are composed of the nanoporous network structure of SiO_2_ particles [6] are the current mainstream direction in the field of aerogel research. Due to their low thermal conductivity of 0.012 to 0.018 W/(m·K) [7], silica aerogels are considered to be the most promising thermal insulation materials. However, pure silica aerogels are characterized by high manufacturing costs, an environmentally unfriendly manufacturing process, low mechanical strength, and high brittleness [8], which greatly limit their large-scale popularization and application [9,10].

The current mainstream method to prepare silica aerogels is to compound them with organic polymer materials to improve their processing properties and broaden their applications. Organic aerogels can be mainly classified into synthetic polymer-based aerogels and biomass-based aerogels in terms of precursors. Synthetic polymer-based aerogels were originally synthesized by Pekala using resorcinol-formaldehyde as raw materials through high-temperature carbonization [11]. The flexible molecular design of polymers makes it easier to control their properties. Unfortunately, most polymers are flammable and produce toxic and carcinogenic substances during combustion [12,13], with side effects from the production stage to the end of their service life [14]. Compared to synthetic polymers, biomass aerogels prepared from naturally occurring renewable materials (polysaccharides, proteins, etc.) have the advantages of low price and wide source [15,16]. At present, studies have shown that biomass materials with C, H, and O as the main components have a certain thermal stability and flame retardancy [17]. For example, the volatile substances produced by the decomposition of sodium alginate at high temperatures are mainly carbon dioxide, and the environmental protection safety is significantly higher than that of synthetic materials. It has been reported that polysaccharides such as Konjac glucomannan (KGM) [17], starch [18], and cyclodextrin [19] have good thermal stability, and can be used to construct new environmentally friendly flame-retardant systems. In addition, some polysaccharides also exhibit good flame-retardant properties, which can partially improve the limiting oxygen index (*LOI*) of materials [20,21].

KGM is a kind of abundant, non-toxic polysaccharide found in konjac tubers [22]. It consists of _D-_mannose and _D_-glucose residues (linked by β-1,4 bonds) in a ratio of 1.6:1 or 1.4:1, depending on the genotype [23,24]. The sugar units of KGM are rich in free hydroxyl groups, which can form an ideal cross-linked structure through hydrogen bonds, and are considered to be a good aerogel framework material [25]. These excellent properties make KGM aerogels and KGM composite aerogels have many broad applications. The mechanical properties and thermal insulation of KGM composite aerogels have been reported in several literatures [26,27,28], but to our knowledge, there has been no research on the relationship between the structure of KGM composite aerogels and their properties. Moreover, it is currently impossible to finely control the structure of aerogels, resulting in poorer thermal insulation and fire-retardant effect than inorganic materials [29,30].

Although there are many studies on inorganic–biomass composite aerogels, they have mainly focused on the comparison of materials used. Several articles have been published on the effect of structural design on the adsorption properties of the material [31], but there are few studies on the thermal and flame-retardant properties of composite aerogels. Physical co-mingling and interpenetrating network (IPN) methods are the most commonly used methods to prepare composite aerogels. It is well known that a flexible three-dimensional network structure is an effective way to overcome the brittle properties of biomass-based aerogels, increase resilience, and reach the required performance [32]. IPNs are three-dimensional network structures composed of two or more interpenetrating polymers with no chemical bonds between them [33,34]. The cross-linking of silica and KGM frameworks in the aerogel networks significantly improves the toughness, while also reducing the porosity and transparency of the samples [35,36,37]. The physical blending method, in which SiO_2_ aerogel particles are mixed and gelled with a second-phase material to form composite aerogels, is a fast, simple, and low-cost classical method for preparing organic–inorganic composite aerogels. The aerogel prepared by the physical blending method can retain the original structure of SiO_2_ aerogels, the technical key is the homogeneous mixing of the components. Compared with simple physical mixing, the cross-linked IPN allows different components to penetrate and entangle with each other. There are some special structural and morphological features such as cell-like structure, interfacial interpenetration, and biphasic continuity for IPNs. These features allow polymers with very different properties or different functions to form stable links with weak interactions (e.g., hydrogen bonding, hydrophobic interactions, van der Waals forces, etc.) and prevent phase separation [38]. This leads to complementary properties or functional synergies between components, such as improving the mechanical strength, thermal insulation properties, and responsiveness of aerogel samples [39].

Therefore, in this work, differences in the structure and properties of aerogel samples prepared by two different preparation methods (simple physical blending and IPN cross-linking) were investigated. The effect of hydrogen-bonding interactions between KGM and silica molecules on their mechanical strength and thermal insulation properties was also explored. First, silica aerogel (SA) microsphere particles were prepared by a hydrolysis–polymerization method using tetraethoxysilane (TEOS) as the silicon source, and KGM–SiO_2_ hybrid composite aerogels (KTB) were prepared by physical blending. Then, the IPN composite aerogels (KTC) were prepared by adding TEOS to the KGM gel by the co-precursor method [40]. The composite aerogels were prepared by compounding KGM and TEOS under two different process conditions. Freeze-drying was chosen to replace supercritical drying due to its safety and low cost [41]. The effects of the addition of different TEOS on the mechanical properties, microstructure, thermal conductivity, and thermal stability of the composite aerogels were explored. In addition, the chemical structure, combination mode, component distribution, and intermolecular interaction of the two composite aerogels were also analyzed. The results show that KTC aerogels have better mechanical properties, thermal insulation, and fire-retardant properties than KTB aerogels, which may be due to the stronger hydrogen-bonding interactions between KGM and silica molecules in KTC aerogels under the unique forcing effect of IPNs. This study attempts to investigate the gelling mechanism of KGM–silica composite aerogels as well as the mechanism of their structural effects on properties at different scales from macroscopic, and microscopic to molecular levels. It is expected to provide a theoretical basis for the design and construction of biomass insulation materials that combine excellent mechanical properties, low thermal conductivity, high safety, and green environmental protection.

## 2. Results and Discussion

### 2.1. Structural Characterization

The SEM images in Figure 1 show that the composite aerogels formed a good three-dimensional network structure, with the KGM macromolecular chains as the skeleton material in the KTB aerogels. The three-dimensional network structure is closely related to the pre-cooling temperature in the first place. The size and distribution of ice crystal growth are strongly related to the nature of the raw material and the pre-cooling temperature. The pre-cooling temperature should be above the freezing point of the sol, the lower the better, to reduce the temperature difference with the freezing temperature, which can more quickly promote ice crystal generation and better control the ice crystal size [42,43]. When the additional amount of SA particles was increased to 1.5%, local agglomeration appeared in the pore wall structure of the aerogel. When the addition of SA was further increased to 2%, more pronounced agglomeration could be seen in the images. As can be seen more clearly in the images, the agglomerated SA is attached to the KGM skeleton, which increased the thickness and local density of the skeleton. The agglomeration may be due to the large specific surface area and high surface energy of the SA microspheres, resulting in SA particles in an energy-unstable state [44] and tending to agglomerate into secondary particles upon stirring. With the addition of SA particles, the distance between SA particles was shortened, and the intermolecular van der Waals force is greater than their gravitational forces, thus making it easier for them to attract each other to agglomerate.

For the KTC aerogels prepared by the co-precursor method, a strong three-dimensional IPN structure was formed. Compared with the K_1_ aerogels, the pore size was reduced, the number of small pores on the pore wall was increased, and the pore channel was reduced, which results in the improvement in mechanical strength. The surface of the aerogel samples with the mass ratio of KGM to SiO_2_ 1:0.5 was uneven, probably due to the small intermolecular force between KGM and silica and the small and disordered pore size. The K_1_TC_0.5_ sample had the lowest concentration of silica; therefore, the resulting silica gel did not fill the entire KGM network. Among all the KTC aerogel samples, the K_1_TC_1.5_ aerogels showed the most uniform pore structure and smooth pore wall. The excessive increase in silica content may lead to the destruction of the three-dimensional network structure of the composite aerogels, as well as the tendency to smaller and more inhomogeneous pore sizes.

The wide range of X-ray diffraction (XRD) patterns in Figure 2A indicates the amorphous nature of the materials. Broad reflections were observed for both composite aerogel samples, and these peaks are typical of sol–gel derived materials from TEOS [45]. The XRD spectrum of the K_1_ aerogel showed two broadened weak peaks near 2θ = 14.86° and 21.56°, indicating that KGM has an amorphous structure [46,47,48].

The FTIR spectra of the KTB and KTC aerogels are shown in Figure 2B,C. The main characteristic absorption peaks of the O-H stretching vibration were at about 3356 cm^−1^, and the main characteristic absorption peaks of the C-H stretching vibration were at about 2883 cm^−1^ [49]. The carbonyl absorption peak at 1640 cm^−1^ was caused by the C=O stretching vibration of the acetyl group in the KGM [50]. The strong absorption peaks at 1025 and 800 cm^−1^ in the spectra were due to the stretching vibration of the organic Si-O-Si and organic Si-C stretching vibration [3]. After the addition of SiO_2_, no new peaks were found in the FTIR spectra of both the KTB and KTC aerogels compared with the K_1_ aerogels, indicating that there was no chemical reaction between TEOS and KGM. For the KGM–SiO_2_ composite aerogels, the wavelengths of the characteristic absorption peaks at about 1025 cm^−1^ and 800 cm^−1^ are enhanced, showing that the cross-linking between KGM and SiO_2_ is formed by a hydrogen bond. With the increase in SA content, the absorption peak of the hydroxyl group in Figure 2B shifted to the lower wavenumber direction, signifying that more hydrogen bonds were formed between the blended molecules. When the addition amount of SA particles was 1%, the absorption peak wavelength of O-H was the shortest, meaning that the cross-linked network was the densest and the hydrogen-bonding force was the strongest. Among the KTC aerogels, K_1_TC_1.5_ has the strongest hydrogen-bonding force. This may be due to the formation of a stronger IPN structure between KGM and SiO_2_. The unique forcing effect of the IPN makes the two molecules tightly intertwined and forms a stronger hydrogen-bond interaction, finally increasing the stability of the sol [51].

The surface chemical composition of the aerogel samples and their oxidation states were further analyzed by X-ray photoelectron spectroscopy (XPS). Carbon signals in plant polysaccharides can generally be classified into four categories, and the C 1s spectrum of KGM in Figure 3B is assigned to four different peaks with binding energies of 289.34, 287.86, 286.46, and 284.84 eV for C=O, C-O-C, C-O, and C-C/C-H, respectively [47,48]. The two additional peaks of Si 2s and Si 2p in the spectra of the K_1_TB_1_ and K_1_TC_1.5_ aerogels indicate that Si was successfully introduced into the composite aerogels.

BET was used to study the effect of different structures on the specific surface area of the aerogels. The nitrogen adsorption and desorption isotherms of the KTB and KTC aerogels are shown in Figure 4A,B. According to the IUPAC classification, the K_1_TB_0.5_ has a type Ⅲ isotherm, which means that the interaction between the molecules in the adsorbate is stronger than the interaction between the adsorbents. The hysteresis loop appears when the adsorption and desorption curves do not overlap with each other. Adsorption metastability and network structure are the main reasons for the appearance of hysteresis loops. There is no adsorption hysteresis loop for the K_1_TB_0.5_. When the additional amount of SA increased to 1% or more, the adsorption–desorption curves changed to type IV isotherms, and capillary aggregation occurred due to the generation of mesopores. Both ends of the curve are convex, and the middle is slightly concave. The curve showed an adsorption hysteresis loop when the relative pressure was between 0.4 and 0.8, and the adsorption rate of nitrogen did not decrease after the adsorption was completed. It is generally acknowledged that the shape of the hysteresis loop is related to the structure of the mesoporous material (e.g., pore size distribution, pore geometry, and connectivity), and IUPAC gives a classification of the hysteresis loop, which is based on de Boer’s earlier classification of hysteresis loops [52].

The physical adsorption isotherm of the KTC aerogels also belongs to the type Ⅳ when KGM and SiO_2_ form an IPN. There is a hysteresis loop caused by capillary condensation in the nitrogen adsorption–desorption curves [53]. With the formation of the SiO_2_ network, the pore structure of the aerogels changes from macropores to mesopores. This also confirmed that the KTC aerogels are a kind of mesoporous material. The average pore size of the composite aerogels was significantly smaller than that of the K_1_, and the total adsorption capacity of the aerogels showed an overall upward trend. The characteristics of the adsorption–desorption curves of all samples were similar to those of the H2-type disordered materials. Moreover, the distribution of the pore size and shape was not well defined, which may be caused by the three-dimensional network structure of the aerogels.

The pore size distribution of the aerogels in Figure 4C,D is derived from the nitrogen adsorption–desorption curves. Table 1 shows that the pore size of the KTC aerogels is smaller and the porosity is higher compared to that of the KTB aerogels. With the increase in silica content, the specific surface area gradually increased, and the pore size showed a trend of first decreasing and then increasing. The test results relying only on BET were not very consistent with the results of the density and porosity tests. The specific surface areas of the KTB aerogels measured by the nitrogen adsorption and desorption curves were all higher than those of the KTC samples, which showed a similar trend to the calculated results of the density, but the smaller pore sizes measured by BET seemed to be inconsistent with the results of the porosity. The main reason may be that the composite aerogel has both subtle mesoporous and nanoporous structures and relatively large macroporous structures (three-dimensional network of polysaccharides) inside, and the BET method is generally more accurate for the determination of mesoporous and microporous materials, but more difficult for macroporous materials. Therefore, to characterize the pore size more accurately, we further selected three aerogels, K_1_, K_1_TB_1_, and K_1_TC_1.5_, to supplement the experiments with the mercury injection method. As also observed by the SEM images, the aerogels also contained large pores inside, and the pore sizes of the K_1_, K_1_TB_1,_ and K_1_TC_1.5_ aerogels were further measured by the mercury injection method in Figure 4E,F. Since the measurement range of the mercury injection method was 5 nm or more, the average pore sizes of the measured aerogels were significantly higher than those measured by the nitrogen adsorption–desorption curves. The average pore size of the K_1_ aerogels measured by the nitrogen adsorption–desorption curve was about 124.34 nm, while the pore size measured by the mercury injection method was about 17.78 µm. The pore diameters of the K_1_TB_1_ and K_1_TC_1.5_ measured by the pressure pump method were also about 32.86 and 17.83 µm, respectively. Therefore, a combination of both methods is needed to analyze the pore size and distribution.

### 2.2. Density, Porosity, and Mechanical Properties

Density and porosity are important parameters that evaluate the porous structure of aerogel materials [54]. Table 1 and Figure 5A,B show the porosity–density line graph of the KTB and KTC aerogels. Compared to the K_1_ aerogels, both the KTB and KTC aerogels showed a trend of increasing density and decreasing porosity with increasing SiO_2_ addition. This should be attributed to the increase in solute content per unit volume of sol with the increase in SiO_2_.

Measuring the mechanical properties of aerogels with a texture analyzer allows an indirect assessment of the driving force for the macromolecular aggregation and the forces maintaining the gel structure in aerogels [55]. Figure 5C presents the results of the mechanical property test for all the prepared KTB samples at a compression ratio of 30%. The compressive capacity of the K_1_ aerogels was only 6.44 kPa without the addition of SA particles. Further analysis showed that a progressively denser cross-linked network existed with the addition of SA particles. The KTB aerogels reached a peak compression capacity of 24.07 kPa when the amount of SA added was equal to that of KGM. Consistent with the results of the nitrogen adsorption and desorption curves, the curve of the KTB aerogels changes from type ΙΙΙ to type ΙV when the SA content reaches 1%. With the continuous addition of SA, the compressive strength showed a decreasing trend, which may be due to the local agglomeration between SA particles and KGM, resulting in an uneven aerogel structure. Overall, these results indicate that the addition of a certain amount of SA can consolidate the pore ratio structure, thereby improving the compressive strength of the aerogel. However, the addition of an excessive amount of SA would destroy the homogeneity of the aerogels and reduce the compressive strength.

However, after the formation of the IPN, the compression resistance of the aerogel was not optimal when the same amount of SiO_2_ and KGM was added. This should be attributed to the weak IPN. As shown in Figure 5D, the stress of the aerogels reached a maximum value of 42.24 kPa when the ratio of KGM to SiO_2_ was 1:1.5. Nevertheless, with increasing SiO_2_ content, the stress presented a decreasing trend. This may be due to the fact that the silica gel has spread over the KGM molecular chain when the concentration of SiO_2_ reaches a high level (1.5 wt%), and the increased SiO_2_ is consumed to reduce the pore size of the aerogel samples (consistent with Figure 1). Combined with the SEM image observation, it is possible that the three-dimensional IPN structure was destroyed and the supporting force became weaker, leading to the decline in the compressive capacity. Meanwhile, the compressive capacity of the K_1_TC_1.5_ aerogels was twice as high as that of the K_1_TB_1_ aerogels (24.07 kPa). The excellent compressive strength of the KTC aerogels should be attributed to the IPN formed between the KGM molecular chains and SiO_2_. In conclusion, the mechanical properties of the KTC aerogels composed of the IPN are significantly improved compared to the KTB aerogels obtained by simple co-blending.

### 2.3. Thermal Insulation Performance and Thermal Stability

There are three basic modes of heat transfer in porous materials: convection, heat conduction, and heat radiation [56]. Aerogel materials mainly rely on solid thermal conductivity (*λ*_s_), gaseous thermal conductivity (*λ*_g_) in pores, and radiative thermal conductivity (*λ*_r_) [57]. *λ*_s_ depends on the aerogel framework, while *λ*_g_ is closely related to the pore structure of the aerogels. Lu et al. proved that *λ*_g_ and *λ*_r_ decrease with increasing density, while the trend of *λ*_s_ was the opposite [58]. The thermal conductivity (*λ*) is closely related to the pore size distribution, pore shape, and pore wall structure of organic–inorganic composite aerogels. Figure 6A,B show the *λ* for the KTB and KTC aerogels with different SiO_2_ contents. Both the KTB and KTC aerogels had an intact and homogeneous appearance and structure, and no significant shrinkage occurred during the freeze-drying process in Figure 6C. With the increasing SiO_2_ content, the *λ* for both aerogels showed a trend of first decreasing and then increasing. This may be due to the increase in *λ*_g_ and decrease in *λ*_s_. The addition of silica disrupted the original closed pore structure of the polysaccharide aerogels, where a significant increase in the open pore structure can be observed in the sem plot (Figure 1D), thus contributing to the increase in *λ*_g_, which is more evident in the KTC aerogels. As the additional amount of silica continued to increase, the microporous wall materials of both composite aerogels became dominated by silica, resulting in a decrease in *λ*_s_.

As shown in Figure 6C, with the addition of SA particles, the *λ* of the aerogels first decreased and then increased, and the minimum value is 0.0646 W/m·K at the K_1_TB_1_. It could be attributed that the addition of mesoporous SA particles complicated the air heat transfer path. Aerogels agglomerate with the amount of SA particles of more than 1%, resulting in porosity and a solid heat transfer increase, leading to an increase in *λ*. Moreover, the *λ* of the KTC aerogels also showed a trend of first decreasing and then increasing, with the lowest value at the K_1_TC_1.5_. Compared to the KTB, the KTC aerogels had a lower *λ* (0.0283 W/m·K), which may be due to the formation of a denser IPN structure between SiO_2_ and KGM. With commercially available inorganic materials (glass wool and rock wool) [14,59], whose thermal conductivity is 0.03–0.04 W/m·K, the optimal aerogel was able to lower the thermal conductivity by around 30% on this basis. In addition, compared to the thermal conductivity of organic conventional materials, such as polyurethane (PUR) [60], extruded polystyrene (XPS) [61], and expanded polystyrene (EPS) [62], the thermal conductivity of the K_1_TC_1.5_ can be reduced by about 20%. The excellent insulation effect was mainly due to the design of the IPN structure and the unique properties of the silica element itself. The low *λ* endowed the biomass-based aerogels with excellent thermal insulation properties so that they have good prospects for application.

To demonstrate the thermal insulation performance of the aerogel samples more visually, the dynamic temperature distribution of the aerogel samples was observed using an infrared camera. As shown in Figure 7, the same volume of aerogel samples was placed on a heating stage at 150 °C and the temperature at the top of the aerogel samples was observed at different times. The temperature of all three aerogels increased with time. However, the temperature of the K_1_TB_1_ prepared by physical blending is 77.3 °C after 30 min, which is a relatively smaller decrease compared to the top temperature of 80.9 °C of the K_1_ aerogels. However, the temperature of the IPN K_1_TC_1.5_ aerogels prepared by the co-precursor method was only 67.8 °C, which was lower than the temperature of the K_1_ aerogel being heated for 10 min. Meanwhile, it was equal to the temperature of the K_1_TB_1_ aerogels being heated for 10 min. Moreover, none of the three samples ended up with significant shrinkage or ignition.

TG and DTG measurements were performed to investigate the thermal stability of several aerogels prepared in this work. Figure 8 shows the TG curves of the K_1_ and KTB aerogels. The sample quality loss can be mainly divided into three stages. All samples showed a first-stage mass loss at about 100 °C, indicating that there was still a small amount of water in the samples. After that, the mass loss of the aerogel samples was mainly in the second stage around 250 °C, probably due to the pyrolysis of the KGM molecular chains. The third stage was the continued weight loss at 400 °C with a small weight loss rate, which may be caused by the oxidative decomposition of methyl groups in the network of the aerogel samples. As the content of SiO_2_ increased, the residual sample mass after TG treatment also increased. The non-decomposable residues of the final composite aerogels are mainly organic carbon and inorganic SiO_2_.

The T_a_ of SA calculated from Figure 8A and Table 2 was 458.292 °C. In addition, the mass loss was only 2.9% as the temperature rises to 800 °C, indicating that SA has good thermal stability. From Table 2, it can be seen that the three temperatures, T_a_ and T_d,_ and DT_d_ of both aerogels varied with increasing silica content. Among them, the KTB aerogels are first decreasing and then increasing, while the KTC aerogels presented the opposite trend. This may be due to the increase in silica in the non-continuous phase, which can firstly disrupt the intermolecular hydrogen bonding of the KGM network while reducing the stability of the aerogel network structure. Moreover, the carbonyl and hydroxyl groups on the surface of KGM are more likely to have an unstable decomposition under heating conditions, leading to a decrease in the three temperatures of T_a_ and T_d_, and DT_d_. In contrast, the formation of the IPN structure of the KTC aerogels promoted the intermolecular interactions of the aerogels and enhanced their structural stability. For the two aerogels with different structures, T_a_, T_d_, and DT_d_ of the K_1_TC_1.5_ were higher than those of the best formulation of the KTB aerogels (K_1_TB_0.5_) (4.9%, 2.7%, and 3.5% higher, respectively), which may be due to the different interaction forces between KGM and silica. The interfacial interpenetration between KGM and silica in the KTC aerogels leads to tighter pinch angles and stronger hydrogen bonds between them, which improves the structural stability, and higher energy potentials are required to break the weak interactions (hydrogen bond) under heating conditions. The higher thermal stability of the K_1_TC_1.5_ was consistent with the thermal conductivity and thermal infrared images are consistent with the results.

### 2.4. Flame-Retardant Performances

The peak heat release rate (PHRR), the temperature corresponding to the arrival of the peak heat release rate (T_PHRR_), the total heat release (THR), and the *LOI* of the prepared aerogels are summarized in Table 3. The higher the HRR, the more heat reaches the surface after combustion, which accelerates the thermolysis of the material. The THR is used to measure the total heat released from the sample during combustion, which mainly depends on the decomposition products. Figure 9 and Table 3 show the combustibility of two different structured aerogels by microscale combustion calorimetry (MCC). The PHRR and THR of both aerogels decreased with increasing SiO_2_ content. However, the PHRR of the KTC aerogels is slightly lower than that of KTB with the same amount of SiO_2_ added, probably due to the formation of the IPN inhibiting the heat release from the aerogels. The HRR of the K_1_TC_1.5_ aerogels was 25.7 W/g, which was 40% of that of the K_1_TB_1_ and 75.43% of that of the K_1_TB_1.5_. Meanwhile, there was no significant difference in the temperature corresponding to the peak heat release rate reached by the aerogel samples, which was maintained at about 320 °C.

The *LOI* values of both the KTB and KTC aerogels were higher than 27%, indicating that the aerogels prepared by these two methods were flame retardant. Compared with the KTB aerogels, the *LOI* of the KTC aerogels was higher and the heat release rate was smaller, indicating its better flame retardancy. The samples prepared by the co-precursor method could form a cross-linked IPN, which reduced the heat transfer of the materials.

Figure 10 shows the macroscopic view of the two formulations of aerogels after combustion in air and the SEM images of the combustion part, respectively. The K_1_ aerogels burned rapidly to powdery ash upon ignition. The results of the combustion experiments showed that the K_1_TB_1_ samples in the combustion part showed a clear tendency to shrink inward (Figure 10A). The corresponding SEM images (Figure 10B) also showed that the polysaccharide skeleton and SA particles clumped together after combustion. This is probably due to the fact that the skeleton structure of the K_1_TB_1_ aerogels is dominated by polysaccharide chains. When the polysaccharides are carbonized at high temperatures, the skeleton structure is disrupted and cannot maintain the microporous structure of the aerogels and agglomerates into clumps (consistent with the SEM images). Figure 10C showed that the network structure of the combustion part of the K_1_TC_1.5_ composite aerogels did not change significantly after combustion. Combined with the SEM image in Figure 10D, the microscopic three-dimensional network structure of the K_1_TC_1.5_ aerogels did not change. This is because the backbone structure of the K_1_TC_1.5_ is composed of polysaccharides and SiO_2_ together. Under combustion conditions, the carbonization of the polysaccharide formed a dense carbon layer along the microporous structure of the aerogels. While the SiO_2_ network was stable in properties and basically does not change, the microporous structure of the aerogels was better to maintain [63]. The results showed that both kinds of composite aerogels can be effective flame retardants. Among them, the KTC with an interpenetrating double network structure can better maintain the original micro and macro structures after high temperature combustion and had better structural stability and flame retardancy. This is consistent with the *LOI* results in Table 3.

## 3. Materials and Methods

### 3.1. Materials

Tetraethoxysilane (TEOS) was purchased from Shanghai Api Chemical Reagent Co., Ltd. (Shanghai, China); ethanol (EtOH) was obtained from Huatian Biotechnology Co., Ltd. (, China); hydrochloric acid (HCl) and sodium hydroxide (NaOH) were supplied by Sinopharm Chemical Reagent Co., Ltd. (Shanghai, China); Konjac glucomannan (KGM, 90.3% glucomannan, *M*_w_ = 5.44 × 10^5^ Da) was supplied by Hubei Qiangsen Konjac Technology Co., Ltd. (Wuhan, China).

### 3.2. Preparation of KTB Aerogels

KGM-based aerogels (KTB and KTC) were prepared by the sol–gel process and freeze-drying method based on previous research [43] with some modifications. The SiO_2_ hydrogels were prepared using a two-step acid-base catalyzed sol–gel process. Transparent silica sol was synthesized via mixing and stirring TEOS, H_2_O, EtOH, and HCl in the ratio proportion of 1:4:1:0.1 at room temperature on a magnetic stirring apparatus at 600 r/min for 12 h. NaOH was dropped into the solution at 1000 r/min to adjust the pH to 8–9, then the gelation process happened and kept for several minutes. Heating at 60 °C for 0.5 h promotes polycondensation. It was dried in an oven and crushed to obtain SA.

KGM was first dissolved in a beaker (100 mL) at 70 °C for 0.5 h. Then, SA particles were added. The mixture was stirred on a stirrer at 500 r/min for 3 h. Subsequently, the sols were infused into molds and placed into the 4 °C refrigerators to age the molds for 0.5 h and immediately frozen in an ultra-low temperature refrigerator at −25 °C for 8 h.

### 3.3. Preparation of KTC Aerogels

The silica sol was first prepared from TEOS, EtOH, H_2_O, and HCl. After hydrolysis at 600 r/min for 12 h, it was added to the KGM gel. After mixing well at 500 r/min for 3 h, NaOH was added to adjust the pH. After a period of dialysis, the samples were poured into molds and aged for 0.5 h in a 4 °C refrigerator and then frozen for 8 h in −25 °C refrigerators. The drying method used in this experiment is freeze-drying (−50 °C and 10 Pa). The KGM addition of all aerogel samples was fixed at 1%, and different samples were prepared by varying the addition of TEOS. The aerogel samples were named K_1_TB_0_ or K_1_TC_0_, and the numbers after the K, TB, and TC are the mass ratios of the two components.

### 3.4. Sample Characterization

The bulk density (*ρ*_b_) of aerogels was determined by the ratio of mass to volume. The density of aerogels can be calculated using the following formula:(1)$$\rho_{b}=\frac{m}{{\left(\frac{D}{2}\right)}^{2}\pi H}$$
where *m* is the dry weight of the aerogels, and *D* and *H* are the diameter and height of the aerogel samples.

The porosity of the aerogels was measured by the ethanol liquid immersion method [64]. The initial mass of the sample was *m*_0_, then it was completely immersed in ethanol and measured as *m*_1_. After a period of vacuum, the sample was taken out and weighed, and the total mass of ethanol and the beaker was recorded as *m*_2_. The porosity of the sample can be calculated as follow:(2)$$P\left(\%\right)=\frac{m_{2}-m_{1}-m_{0}}{m_{2}-m_{1}}\times 100\%$$

The specific surface area (*S*_BET_) was obtained through nitrogen adsorption–desorption and the Brunauer–Emmet–Teller (BET) model (BET, AUTOSORB-1MP, Quantachrom, Boynton Beach, FL, US). The pore size and distribution of the aerogels were determined using nitrogen adsorption–desorption curves, BET models, and high-performance fully automated injection mercury instruments (Micromeritics AutoPore IV 9500). The samples were first dried in an oven at 80 °C for 24 h. The test pressure was first gradually increased from low pressure to 60,000 psi and then slowly decreased to 14.7 psi.

The microstructure of the aerogels was observed via scanning electron microscopy (SEM) (JSM6390LV, JEOL, Tokyo, Japan) at a magnification of ×50 and ×200. The samples were cut into 5 mm × 5 mm × 1 mm circular pieces using a sharp razor blade.

A high-resolution X-ray diffractometer (Empyrean, Dordrecht, The Netherlands) was used for the diffraction analysis of the aerogels with a scan rate of 5°/min and a 2θ range of 5–50°.

Fourier transform infrared spectroscopy (FTIR) was used to analyze the information of functional groups using a NEXUS (England) in the wavenumber range 600 to 4000 cm^−1^.

The surface elements were validated by X-ray photoelectron spectroscopy (XPS, PHI5000 VersaprobeI, Japan). CasaXPS software was used to process the data.

The mechanical properties of the prepared aerogels were tested by a TMS-PRO texture analyzer (TA. XT Plus, Stable Micro Systems, Surrey, UK), and samples were equilibrated for 48 h at 40 °C drying. The compression rate of the probe was 0.5 mm/s, and the compression ratio was 30%. The compressive strength and elasticity were obtained by secondary compression. Stress (*σ*) was calculated using the following standard equations:(3)$$\sigma =\frac{F}{S}$$
where *F* is the force (in N) applied on the sample surface, and *S* (in mm^2^) is the contact area between the probe and the sample.

The thermal conductivity of the aerogel samples was recorded at room temperature by a thermal conductivity tester (DRPL-2A, Xiangtan Instrument Co., Ltd., Xiangtan, China).

The aerogel samples were fixed between the heat source (150 °C heating table) and the thermal imager at the same distance for the heat resistance test. The thermographic images were recorded by an infrared thermal camera (323Pro, FOTRIC Ltd., Shanghai, China). The analysis software was used to process the data.

Thermal stability was conducted by using thermogravimetric (TG) and derivative thermogravimetric (DTG) analysis. With the nitrogen flow rate of 30 cm^3^/min, the aerogels were heated from room temperature to 800 °C at a heating rate of 5 °C/min in an N_2_ atmosphere, and the weight loss curve was recorded.

The fire-retardant property of the aerogel samples was determined by a microscale combustion calorimeter (MCC, FTT0001, FTT Ltd., West Sussex, UK).

The limiting oxygen index (*LOI*) was measured by a CH-2CZ oxygen index tester (Nanjing Shangyuan Analysis Instrument Company, Nanjing, China). The samples were cut into strips of 80 mm × 4 mm × 10 mm for testing. Then, the residual components that were completely burned were taken to observe their microstructures under scanning electron microscopy. The *LOI* was calculated using the following standard equations:(4)$$LOI=\frac{C_{0}}{C_{0}+C_{N}}$$

## 4. Conclusions

In this work, two kinds of inorganic–biomass composite aerogels were prepared, and their structural analysis, thermal insulation, and fire-retardant properties were carried out. The KTC aerogels prepared by the co-precursor of KGM and TEOS not only have strong mechanical properties but also have a thermal conductivity of 0.028–0.036 W/m·K, which is reduced to 1/3 of the thermal conductivity of 0.065–0.095 W/m·K compared with that of the aerogels prepared via the physical blending of KGM and SA particles. The peak decomposition temperature of the KTC aerogels was 30 °C higher than that of the KTB due to the IPN of internal cross-linking that prevents heat transfer. Data from HRR and THR show that the cross-linked network formed by KGM and TEOS has a better enhancement of its flame-retardant performance. Optimizing the microstructure of aerogels in the preparation method, can first significantly improve the thermal insulation performance and have a good flame-retardant effect. This work provides new ideas for the preparation of aerogels with thermal insulation and flame-retardant properties. Further investigations are planned to improve their mechanical properties for wider applications.

## Figures and Tables

**Figure 1 molecules-28-01691-f001:**
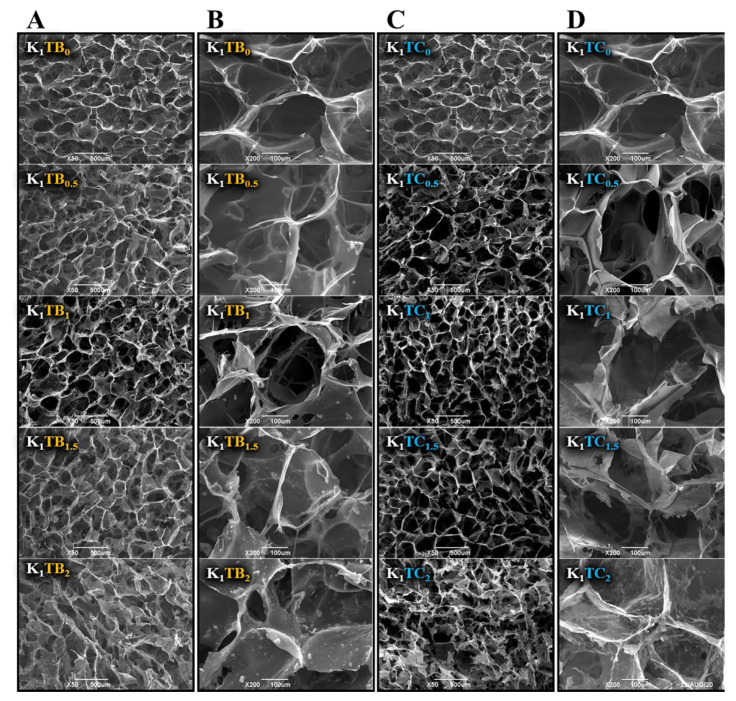
The microscopic morphology of KTB aerogels ((**A**) at 50 times; (**B**) at 200 times), and KTC aerogels ((**C**) at 50 times; (**D**) at 200 times).

**Figure 2 molecules-28-01691-f002:**
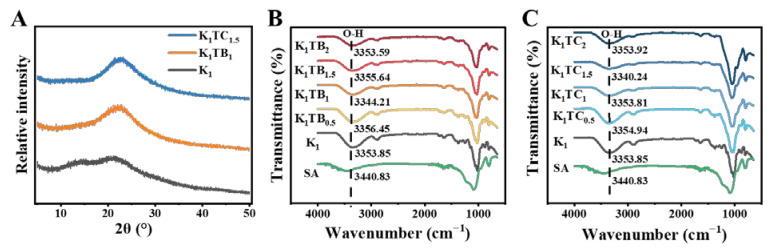
XRD spectra of aerogels (**A**) and FTIR spectra of KTB aerogels (**B**) and KTC aerogels (**C**).

**Figure 3 molecules-28-01691-f003:**
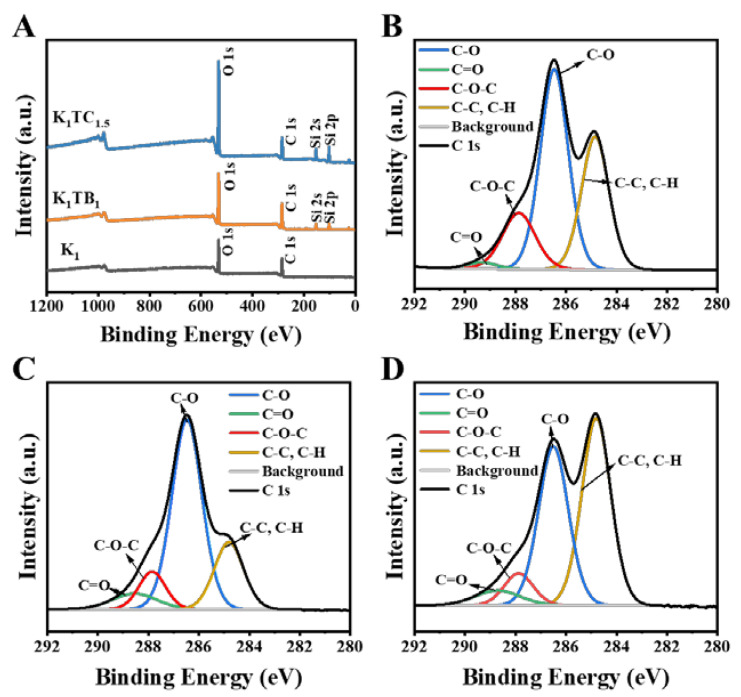
XPS spectra of K_1_, K_1_TB_1_, and K1TC_1.5_ aerogels (**A**) and XPS spectra in the C 1s region for K_1_ aerogels (**B**), K_1_TB_1_ aerogels (**C**), and K_1_TC_1.5_ aerogels (**D**).

**Figure 4 molecules-28-01691-f004:**
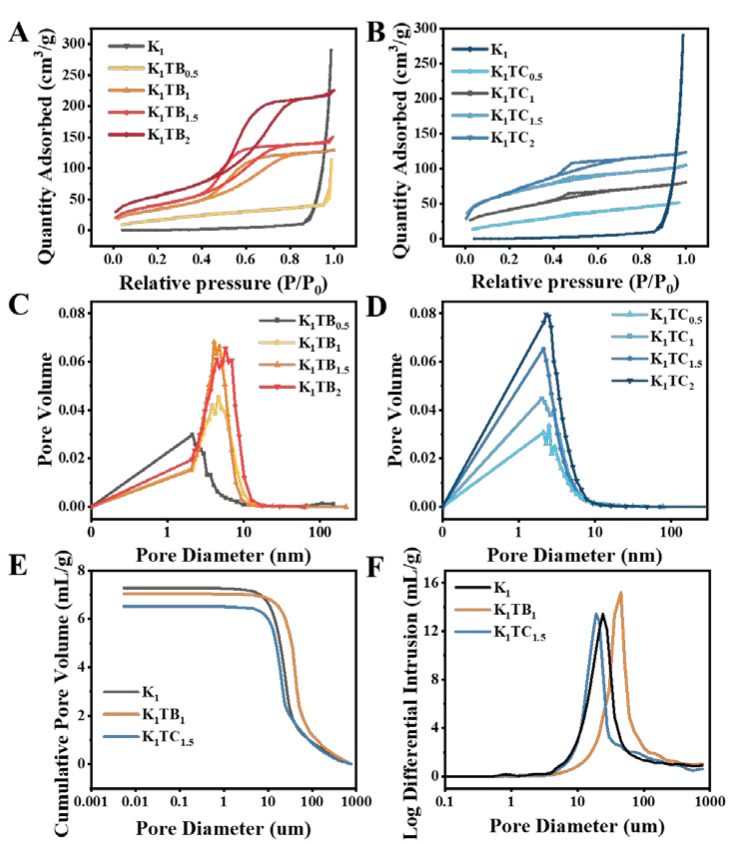
Nitrogen adsorption–desorption curves (**A**,**B**) and pore size distribution curves (**C**,**D**) of K_1_, KTB aerogels, and KTC aerogels and cumulative pore volume and log differential intrusion of K_1_, K_1_TB_1_, and K_1_TC_1.5_ aerogels (**E**,**F**) measured by mercury injection method.

**Figure 5 molecules-28-01691-f005:**
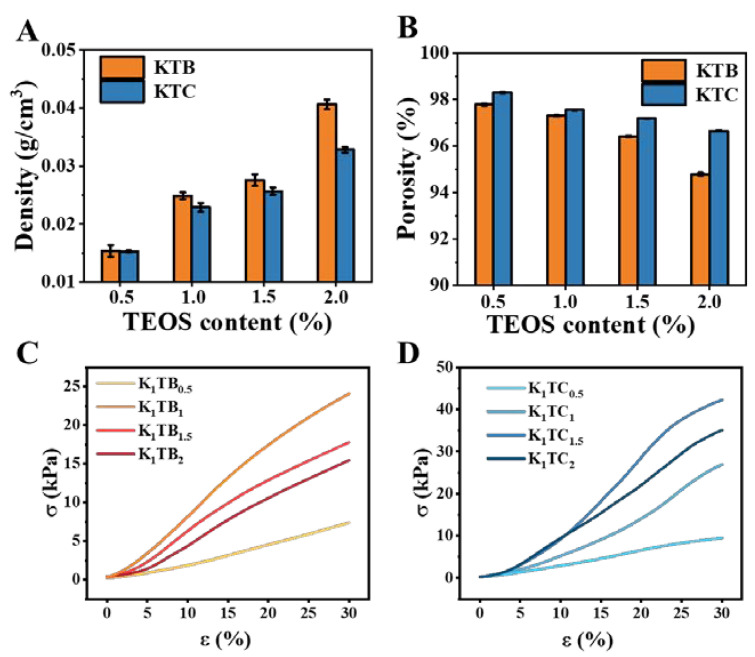
The density (**A**) and porosity (**B**) of KTB and KTC aerogels, and stress–strain curves for KTB aerogels (**C**) and KTC aerogels (**D**).

**Figure 6 molecules-28-01691-f006:**
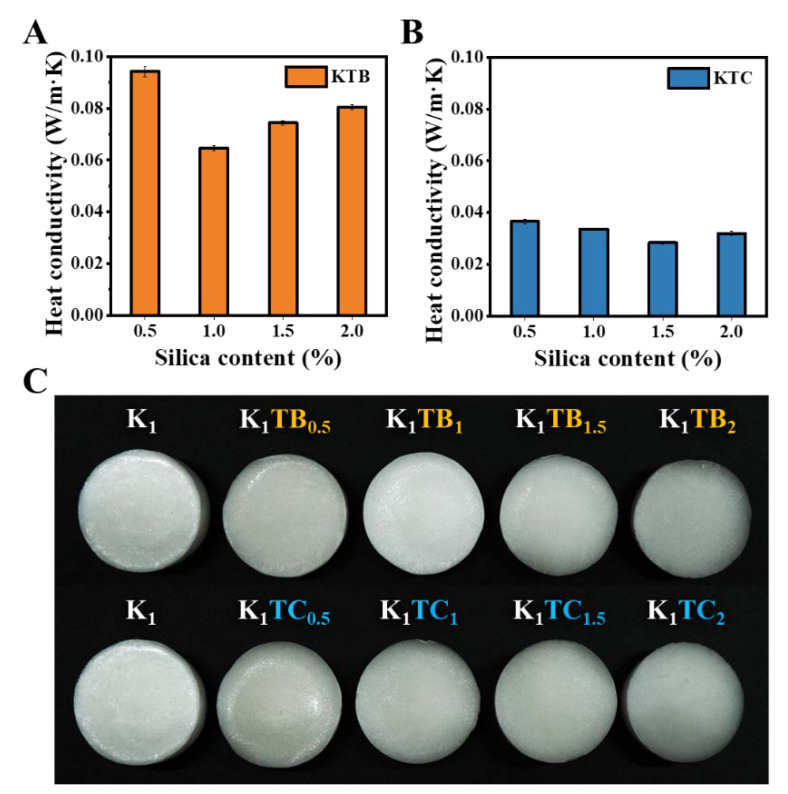
Thermal conductivities of KTB aerogels (**A**) and KTC aerogels (**B**), and macrographs of K_1_, KTB, and KTC aerogels with a diameter of 4 cm (**C**).

**Figure 7 molecules-28-01691-f007:**
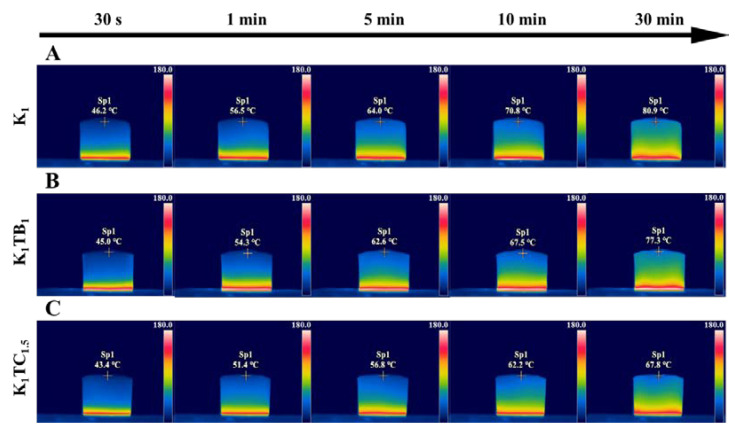
Thermal infrared images of K_1_ (**A**), K_1_TB_1_ (**B**), and K_1_TC_1.5_ (**C**) in 30 min.

**Figure 8 molecules-28-01691-f008:**
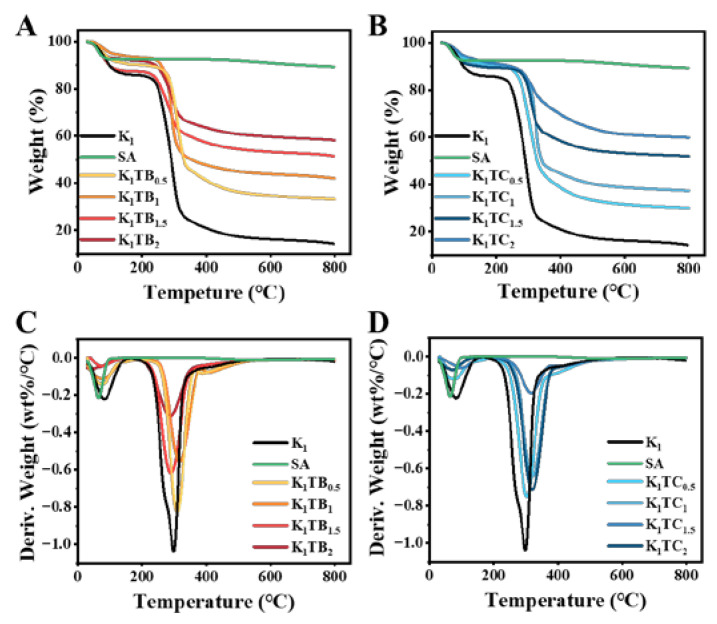
TG (**A**,**B**) and DTG (**C**,**D**) curves of KTB aerogels and KTC aerogels.

**Figure 9 molecules-28-01691-f009:**
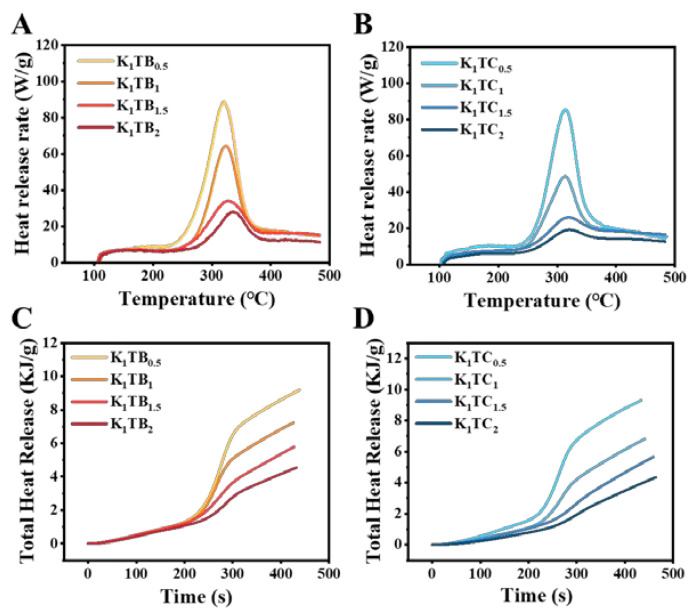
Heat release rate (HRR, **A**,**B**) and total heat release (THR, **C**,**D**) plots of KTB and KTC aerogels.

**Figure 10 molecules-28-01691-f010:**
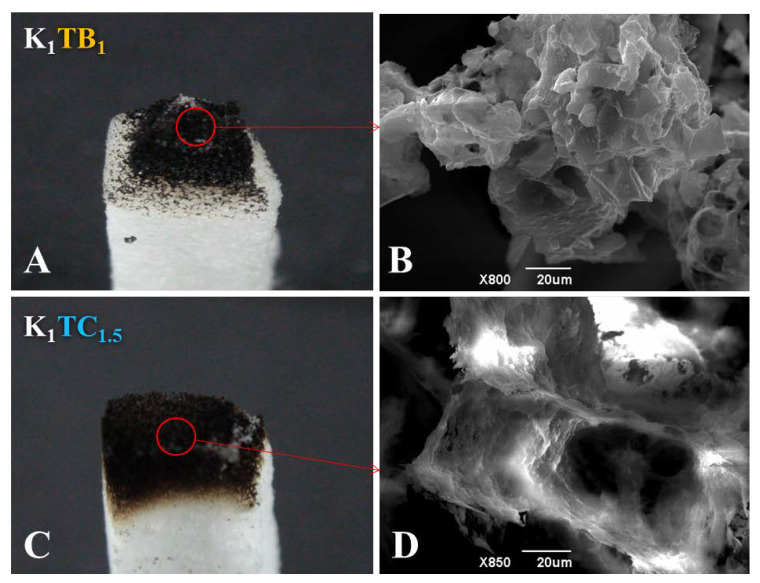
(**A**,**C**) Macroscopic photograph and (**B**,**D**) microscopic morphology of (**A**,**B**) K_1_TB_1_ and (**C**,**D**) K_1_TC_1.5_ aerogels after combustion.

**Table 1 molecules-28-01691-t001:** Basic physical parameters of KTB and KTC aerogel samples.

Samples	*S*_BET_ (m^2^/g)	Pore Size (nm)	Porosity (%)	Density (g/cm^3^)
K_1_	43.437	124.388	98.55	0.0106
K_1_TB_0.5_	86.086	10.344	97.78	0.0154
K_1_TB_1_	192.341	6.309	97.31	0.0249
K_1_TB_1.5_	228.337	6.156	96.42	0.0276
K_1_TB_2_	291.871	6.851	94.79	0.0406
K_1_TC_0.5_	131.588	3.316	98.30	0.0153
K_1_TC_1_	167.225	2.882	97.55	0.0229
K_1_TC_1.5_	180.694	3.676	97.19	0.0257
K_1_TC_2_	232.868	3.103	96.65	0.0328

**Table 2 molecules-28-01691-t002:** The initial decomposition temperature T_a_ (°C), maximum decomposition temperature T_d_ (°C), maximum decomposition temperature by DTG DT_d_ (°C), and degradation mass DM (%) calculated from the TG and DTG curves.

Samples	T_a_ (°C)	T_d_ (°C)	DT_d_ (°C)	DM(%)
K_1_	248.962	323.511	296.053	71.47
SA	458.292	/	/	2.91
K_1_TB_0.5_	274.338	336.779	309.5	57.01
K_1_TB_1_	270.436	319.495	317.333	52.12
K_1_TB_1.5_	246.517	326.129	290.167	36.18
K_1_TB_2_	255.771	331.542	287.833	33.97
K_1_TC_0.5_	272.007	344.636	303.667	60.18
K_1_TC_1_	283.530	343.239	320.5	52.33
K_1_TC_1.5_	288.069	346.032	320.56	37.94
K_1_TC_2_	275.849	375.538	320.167	31.52

**Table 3 molecules-28-01691-t003:** The flame retardancy of aerogel samples from limiting oxygen index (*LOI*) and microscale combustion calorimeter (MCC) tests.

Samples	PHRR (W/g)	T_PHRR_ (°C)	THR (KJ/g)	*LOI* (%)
K_1_TB_0.5_	88.7	320.7	9.1	27.2
K_1_TB_1_	64.3	333.5	7.2	27.3
K_1_TB_1.5_	34.2	326.2	5.7	28.0
K_1_TB_2_	27.9	336.4	4.5	28.5
K_1_TC_0.5_	85.2	313.7	9.3	34.7
K_1_TC_1_	48.6	316.3	6.8	37.1
K_1_TC_1.5_	25.8	323.7	5.6	38.2
K_1_TC_2_	19.0	318.9	4.3	38.5

## Data Availability

The data presented in this study are available on request from the corresponding author.

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
