# Peer review of "Study on the Influence of the Preparation Method of Konjac Glucomannan-Silica Aerogels on the Microstructure, Thermal Insulation, and Flame-Retardant Properties"

_molecules, 2023, doi:10.3390/molecules28041691_

Round 1
Reviewer 1 Report
The authors reported the study on the KGM-silica composite aerogels. Two different KGM-silica aerogels were synthesized to investigate the differences in structures and performances. It is an interesting work about navel aerogel development However, it cannot be accepted for publication after reviewing the whole manuscript owing to the following issues:
1) This work aimed to improve the mechanical properties of silica aerogel by incorporating KGM and silica components. However, it seems that the resulting aerogels are characteristic of KGM aerogel, which has totally different structure and performances comparing with silica-based aerogels, such as colloidal microstructure. Therefore, the manuscript should be reorganized to clarify the issues.
2) Two different KGM-silica aerogels with physical blending and co-precursor methods were synthesized to compare their differences. Strictly speaking, it seems that the two samples do not have comparability. KTB samples were prepared with silica xerogel which was obtained after drying in an oven. KTC samples were obtained with freeze drying. Not only the composite method, the drying method affect the structure of aerogel significantly. Ambient pressure drying could result in critical structure failure of silica aerogel. Even if two samples had difference in composite method, the fundamental difference should be presented and proved, such as the chemical structure (bonding style), silica particle distribution, the evidence of cross-linking, etc. It seems that there is no qualitative difference between KTB and KTC from the presented data.
3) More details about the sample synthesis and characterization should be provided. For example, the freeze-drying temperature and pressure of KTB, the drying temperature of SA and its structure information, the solvent of KGM for KTB preparation, the details of KGM glue, dialysis? The freezing point of ethanol is much lower than -25C, the sample preprocessing? What is K1, pure KGM aerogel? check the porosity test. Pore size distribution derived from N2 adsorption/desorption isotherms should be provided. For KTB, how to control the component of the composite aerogel. For KTC, 1% KGM content, weight fraction? The scaleplate or reference substance should be involved in the digital photographs of samples
4) Concrete embodiment and discussions about the relationship between the structure of KGM composite aerogels and their properties
5) The structure difference of different samples should be discussed, such as specific surface area and pore volume from N2 adsorption test. Obviously, there are a large amount of macropores in the aerogels, mercury injection method should be conducted to analyze the whole pore structure in combination with N2 adsorption method. The pore size in Table 1 is unbefitting obviously.
6) W/(mK), W/(m·K); other grammar mistakes. Aerogels … it…
7) Moreover, it is currently impossible to finely control the structure of aerogels, resulting in their worse thermal insulation and fire-retardant effect compared to inorganic materials. According to the context, the structure of aerogels should be correct to the structure of KGM aerogels. All other similar parts should be corrected.
Author Response
- This work aimed to improve the mechanical properties of silica aerogel by incorporating KGM and silica components. However, it seems that the resulting aerogels are characteristic of KGM aerogel, which has totally different structure and performances with silica-based aerogels, such as colloidal microstructure. Therefore, the manuscript should be reorganized to clarify the issues.
Reply: Thanks very much for the suggestion. I am sorry for the confusion caused by the oversimplification presented in the introduction. In fact, the KTB aerogels prepared by physical blending retain part of the structure and properties of silica aerogels, as proved by the results of FTIR, TG, etc. This has been supplemented in the manuscript with relevant data from SA. The manuscript has been reorganized to clarify the issues and clarified to illustrate the reasons for choosing KTB and KTC aerogels for comparative studies and the differences that exist in their structures and properties.
- Two different KGM-silica aerogels with physical blending and co-precursor methods were synthesized to compare their differences. Strictly speaking, it seems that the two samples do not have comparability. KTB samples were prepared with silica xerogel which was obtained after drying in an oven. KTC samples were obtained with freeze drying. Not only the composite method, the drying method affect the structure of aerogel significantly. Ambient pressure drying could result in critical structure failure of silica aerogel. Even if two samples had difference in composite method, the fundamental difference should be presented and proved, such as the chemical structure (bonding style), silica particle distribution, the evidence of cross-linking, etc. It seems that there is no qualitative difference between KTB and KTC from the presented data.
Reply: Thanks for the suggestion. Actually, both KTB and KTC aerogels were prepared by freeze-drying. Only the SA particles were prepared by using a two-step acid-base catalyzed sol-gel process and dried via an oven to retain the structure of silica aerogels. Compared with simple physical mixing, cross-linked interpenetrating networks (IPN) allow different components to penetrate and entangle with each other. The special structural and morphological features such as cell-like structure, interfacial interpenetration, biphasic continuity, etc allow polymers with very different properties or different functions to form stable links with weak interactions (e.g. hydrogen bonding, hydrophobic interactions, van der Waals forces, etc.) and prevent phase separation (Dhand, A. P.; Galarraga, J. H.; Burdick, J. A. Trends in Biotechnology 2021, 39, 519-538.), thus achieving complementary properties or functional synergies between components, such as improving the mechanical strength, thermal insulation properties and responsiveness of aerogel samples (Bajpai, A. K.; Shukla, S. K.; Bhanu, S.; Kankane, S. Progress in Polymer Science 2008, 33, 1088-1118.). Therefore in this paper, we mainly investigated the effect of different combinations between KGM and SiO2 on the properties of composite aerogels. By comparing their mechanical strength and thermal insulation properties, it can be seen that the KTC aerogels prepared by the co-precursor method have better properties than the KTB aerogels prepared by the physical mixing method. Although both aerogels are physically cross-linked, there exist major differences in their properties. which may be due to the stronger hydrogen bonding interactions between KGM and silica molecules in KTC aerogels under the unique forcing effect of IPN.
- More details about the sample synthesis and characterization should be provided. For example, the freeze-drying temperature and pressure of KTB, the drying temperature of SA and its structure information, the solvent of KGM for KTB preparation, the details of KGM glue, dialysis? The freezing point of ethanol is much lower than -25C, the sample preprocessing? What is K1, pure KGM aerogel? check the porosity test. Pore size distribution derived from N2 adsorption/desorption isotherms should be provided. For KTB, how to control the component of the composite aerogel. For KTC, 1% KGM content, weight fraction? The scaleplate or reference substance should be involved in the digital photographs of samples.
Reply: Thanks! K1 aerogels are pure KGM aerogels. The dialysis we undertake before freezing is to remove the ethanol and ions from the gum solution, so the final sample before freeze-drying was free of ethanol. KGM and SiO2 content are mass fractions. Freeze drying is carried out at -50 ℃ and 10 Pa. Pore size distribution curves derived from nitrogen adsorption-desorption isotherms have been provided and analyzed (Figure 4C and D). The samples were all prepared in six-well plates of the same size with wells of 4 cm in width and 1.5 cm in depth.
- Concrete embodiment and discussions about the relationship between the structure of KGM composite aerogels and their properties.
Reply: Thank you very much for your valuable suggestions. A detailed discussion about the relationship between the structure of KGM composite aerogels and their properties has been supplemented in the manuscript. FTIR spectra showed that the O-H absorption peak of the optimally formulated KTC aerogels (K1TC1.5) is red-shifted compared to that of the optimally formulated KTB aerogel (K1TC1), indicating that the hydrogen bonding between components of KTC aerogel with IPN structure was enhanced. The surface chemical composition of the aerogels and their oxidation states were further analyzed by XPS. The pore size distribution of the aerogels was further refined and analyzed by supplementing some of the BET data as well as the piezometric test data. Property evaluation results showed that KTC aerogels have better mechanical properties, thermal insulation and fire-retardant properties than KTB aerogels, which may be due to the unique forcing effect of IPN, the tight entanglement between KGM and silica molecules in KTC aerogels and stronger hydrogen bonding interactions prompting the formation of stable bonds, resulting in complementary properties and functional synergies between the components.
- The structure difference of different samples should be discussed, such as specific surface area and pore volume from N2 adsorption test. Obviously, there are a large amount of macropores in the aerogels, mercury injection method should be conducted to analyze the whole pore structure in combination with N2 adsorption method. The pore size in Table 1 is unbefitting obviously.
Reply: Thanks! It is obvious from the SEM image that the network structure had large pores. Due to the limitation of the nitrogen adsorption method, the pore size distribution should be analyzed in combination with the mercury injection method. Therefore, we have used the mercury injection method to analyze the pore size of K1, K1TB1, and K1TC1.5 aerogels. The average pore size was significantly higher than that derived from the nitrogen adsorption-desorption curves. The pore size and distribution should be fully analyzed by the nitrogen adsorption method together with the mercury injection method.
- W/(mK), W/(m·K); other grammar mistakes. Aerogels … it…
Reply: Thanks for the suggestion. We have tried our best to improve the language. Some spelling and grammatical errors in the manuscript have been corrected.
- Moreover, it is currently impossible to finely control the structure of aerogels, resulting in their worse thermal insulation and fire-retardant effect compared to inorganic materials. According to the context, the structure of aerogels should be correct to the structure of KGM aerogels. All other similar parts should be corrected.
Reply: Thanks very much for the suggestion. Although aerogels have had a deep history of research, there is currently no way to control their network structure, making some of their properties inferior to those of inorganic materials.
Reviewer 2 Report
The present manuscript entitled "Study on the influence of the preparation method of Konjac glucomannan-silica aerogels on the microstructure, thermal in-sulation, and flame retardant properties " examines the effect of microstructure on the properties of aerogel materials, two aerogels with different structures were prepared using Konjac glucomannan (KGM) and tetraethoxysilane (TEOS) via physical blending (KTB) and co-precursor methods (KTC), respectively. The structural differences between the KTB and KTC aerogels were characterized and the thermal insulation and fire-retard-ant properties were further investigated in this study. These data are interesting however, I am providing some minor comments (related to the Introduction and the results & discussion) and then a list of these comments that need to be addressed. Please find below these comments. Overall, the quality of this paper is suitable for publication in molecules in terms of presentation, content, and description.
1. The introduction reads nice and adequate but the motivation part not enough. Authors should provide a clear explanation about the reason for choosing KTC and KCB aerogels with references.
2. It is suggested to include the XRD planes with the JCPDS card no. in K1 XRD it seems to be two plans are appeared for the clear understanding of the readers can mention the plans clearly.
3. It is recommended to include the XPS spectra of the optimum aerogels (SA) product.
4. It is suggested that the author can analyze it more deeply and clarify the impacts of SA in TG, DSC studies.
Author Response
- The introduction reads nice and adequate but the motivation part not enough. Authors should provide a clear explanation about the reason for choosing KTC and KCB aerogels with references.
Reply: Thanks. Your advice is very helpful. The reason for choosing KTC and KCB aerogels is to comparatively study the mechanisms of intermolecular interactions, structures at different scales and the different effects on properties between the interpenetrating network (IPN) aerogels and aerogels prepared by simple physical blend. As well as to explore the synergistic mechanisms of intermolecular interactions between polysaccharides and silica hydroxyl groups on their complementary properties or functions, to simultaneously improve the thermal insulation, mechanical properties and safety of aerogels. Paragraphs 3 and 4 of the first part (introduction) of the original text have been substantially revised and added to further explain the reasons for choosing KTB and KTC for the study, and to reorganize the writing to highlight the objectives of the study.
- It is suggested to include the XRD planes with the JCPDS card no. in K1 XRD it seems to be two plans are appeared for the clear understanding of the readers can mention the plans clearly.
Reply: Thanks for the suggestion. The two planes in which the K1 aerogel appears have been explained accordingly in the revised manuscript.
- It is recommended to include the XPS spectra of the optimum aerogels (SA) product.
Reply: Thank you for your comments. The XPS spectra of K1, K1TB1 and K1TC1.5 have been added to the revised manuscript (Figure 3A and B). And corresponding results and discussions were added to the revised manuscript.
- It is suggested that the author can analyze it more deeply and clarify the impacts of SA in TG, DSC studies.
Reply: Thanks. Silica has a significant effect on the enhancement of the thermal stability of composite aerogels, the Thermogravimetric (TG) and Derivative Thermogravimetric (DTG) which have been analyzed in detail and in-depth in the revised manuscript.
Reviewer 3 Report
Attached please find the comments.

Author Response
Dear Prof. Zhang and the reviewers,
Many thanks for your email regarding the reviewers’ comments on our paper submitted to Molecules.
Title: Study on the influence of the preparation method of Konjac glucomannan-silica aerogels on the microstructure, thermal insulation, and flame retardant properties
Manuscript Number: molecules-2099177
Authors: C. Li et al.
We much appreciate the reviewers, who did give us valuable comments to improve the paper. This manuscript was revised thoroughly according to the comments and minor revisions were made as listed below.
Please do not hesitate to contact me should you have any questions. Thank you very much and looking forward to your kind consideration.
Best wishes,
Cao Li, Prof., Ph. D.
School of Health Science and Engineering, Hubei University, Wuhan 430062, China.
Email: [email protected]
Reply to the comments:
Reviewer #1:
- The stirring speed and stirring time have a very critical effect on the preparation of aerogels, but the optimal preparation conditions are not reflected in the article.
Reply: Thanks very much for the suggestion. The conditions for the preparation of aerogels were based on previous studies. There are three steps in the preparation of the samples involving stirring: the addition of acid, the addition of base, and co-mixing with KGM.
The stirring rate and time for each stage have been added to the manuscript (Section 2.2 and 2.3).
- Please analyze the effect of high and low pre-cooling temperature on the aerogel by microstructure and how to choose the best pre-cooling temperature.
Reply: Thanks for the suggestion. The pre-cooling temperature was directly applied to the team's prior experience conditions, so no additional optimization experiments were conducted for the pre-cooling temperature, but there are supplemental explanations for the effect of precooling temperature on the aerogel structure in the manuscript (Line 3-9, Page 7).
- Please explain the principle of the application of composite aerogel as a thermal insulation material and compare it with other common insulation materials to show its advantages.
Reply: Thanks! First of all, aerogels have high porosity and small pore size, therefore reducing the thermal conductivity of the gas phase (λg) inside the material. Also with its low density and complex three-dimensional network structure, the solid skeleton structure prolongs the heat transfer path, which consequently reduces the solid thermal conductivity (λs). These are the main reasons why composite aerogel can reduce thermal conductivity. We have compared the aerogel samples with commercially available inorganic materials and organic conventional materials in the paper to supplement and analyze the reasons for their excellent thermal insulation properties (Line 8-15, Page 14).
- For the graphs where the results need to be compared, please try to compare them under the same angle. Some graphs are too small for easy access to the results, so a reasonable graph layout is needed.
Reply: Thank you very much for your valuable suggestions. We have adjusted the size of the text in the charts and icons accordingly and reorganized the graphic layout.
- What is the difference between the results of density and porosity tests of composite aerogels and BET tests described in the text, and give the basis for the choice.
Reply: Thanks! The results of density and porosity are not very consistent with those of BET. Therefore, to more accurately characterize the pore structure of aerogels, we further selected K1, K1TB1, and K1TC1.5 to refine the experiments by mercury injection, and the results showed that K1TC1.5 had a smaller pore size distribution compared to K1 and K1TB1. The concrete analysis has been explained in the manuscript (Line 10-24, Page 11).
- For the description of the conclusion you can add what advantages the previous results bring compared to the results of your own design.
Reply: Thanks for the suggestion. We have refined the conclusion to reflect the advantages of the article (Line 10-12, Page 21), and the improvement of the preparation method can effectively optimize its structure and thus improve its thermal insulation and flame retardant effect, providing a certain reference value for subsequent research.
- Some of the images in TG and DTG are so light in line color selection and background color that they are not easily distinguishable, and please explain the effect of different materials and scale addition on the structure through these two images.
Reply: Thanks very much for the suggestion. The colors of inappropriate curves in the image have been adjusted to distinguish them. Table 2 shows a summary of the TG and DTG curves. The Ta, Td, and DTd (maximum decomposition temperature by DTG) of KTC aerogels are apparently higher than that of KTB aerogels, probably due to the formation of its internal interpenetrating network structure which enhances the stability of its structure. We have refined and supplemented the original article with an analysis of the effect of different materials and additions on the structure.
- Some sentences are quite long and lack proper structure and punctuation. Please proofread the manuscript again thoroughly and pay attention to the English expression.
Reply: Thanks! The sentences that are too long and incomplete in structure have been corrected in the manuscript. And the entire text has been thoroughly checked for words, grammar, and expressions.
- Please use the SEM of the sample before and after combustion to reveal the flame retardant properties as a way to reveal which method of preparing the material has better thermal insulation properties.
Reply: Thanks for the suggestion. The main paper has been supplemented at the end with sample pictures and microscopic morphology of K1TB1 and K1TC1.5 aerogels after combustion (Figure 10). And the reasons for the difference in the results of these two aerogels after combustion have been preliminarily analyzed. It is clear from the graph that K1TC1.5 is more effective in flame retardation, probably due to the interpenetrating network structure formed can effectively prevent heat transfer.
- The references should be expanded. Some new literatures might be help the authors to further deepen the understanding of reaction mechanism as well as newest developing in this field (Journal of Environmental Management, 2023, 326: 116790 Regeneration mechanism of a novel high-performance biochar mercury adsorbent directionally modified by multimetal multilayer loading).
Reply: Many thanks to the reviewer for the references. We have expanded the references and this latest article is innovative in terms of material structure design and adsorption performance and reveals the mechanism of mercury removal and regeneration. It is a valuable guide to the structural design of aerogel in this paper and is so enlightening that it has been included in the references.